# Preliminary Study: Proteomic Profiling Uncovers Potential Proteins for Biomonitoring Equine Melanocytic Neoplasm

**DOI:** 10.3390/ani11071913

**Published:** 2021-06-27

**Authors:** Parichart Tesena, Amornthep Kingkaw, Wanwipa Vongsangnak, Surakiet Pitikarn, Narumon Phaonakrop, Sittiruk Roytrakul, Attawit Kovitvadhi

**Affiliations:** 1Graduate Student in Animal Health and Biomedical Science Program, Faculty of Veterinary Medicine, Kasetsart University, Bangkok 10900, Thailand; parichart.tes@mahidol.ac.th; 2Department of Clinical Science and Public Health, Faculty of Veterinary Science, Mahidol University, Salaya, Puttamonthon, Nakhon Pathom 73170, Thailand; 3Interdisciplinary Graduate Program in Bioscience, Faculty of Science, Kasetsart University, Bangkok 10900, Thailand; amornthep.ki@ku.th; 4Department of Zoology, Faculty of Sciences, Kasetsart University, Bangkok 10900, Thailand; fsciwpv@ku.ac.th; 5Omics Center for Agriculture, Bioresources, Food, and Health, Kasetsart University (OmiKU), Bangkok 10900, Thailand; 6Genetic Engineering and Bioinformatics Program, Graduate School, Kasetsart University, Bangkok 10900, Thailand; surakiet.pi@ku.th; 7Functional Ingredients and Food Innovation Research Group, National Center for Genetic Engineering and Biotechnology, National Science and Technology Development Agency, Pathum Thani 12120, Thailand; narumon.pha@biotec.or.th; 8Department of Physiology, Faculty of Veterinary Medicine, Kasetsart University, Bangkok 10900, Thailand

**Keywords:** bioinformatics, equine melanocytic neoplasm, grey horse, potential proteins, serum proteomics

## Abstract

**Simple Summary:**

Equine melanocytic neoplasm (EMN) affects grey horse worldwide, with the highest prevalence of approximately 80% in older grey horses. However, the efficiencies of both therapeutic and prevention strategies are not high. We investigated a less invasive technique to detect EMN using proteomics via serum protein expression. The serum sample can be used for proteomics analysis to identify and quantify differences in the expression of proteins between normal and EMN grey horses. In addition, it can detect the expressed patterns of proteins and relevant pathophysiology pathways of EMN. We summarised the differential protein expression in each stage, with the overall aim to identify potential proteins in the early stage of EMN and the severity stage of EMN. The potential proteins could be used for biomonitoring EMN, facilitating prevention and treatment based on the preliminary results in this study.

**Abstract:**

Equine melanocytic neoplasm (EMN) is a cutaneous neoplasm and is mostly observed in aged grey horses. This preliminary study aimed to identify potential proteins to differentiate normal, mild and severe EMN from serum proteomic profiling. Serum samples were collected from 25 grey horses assigned to three groups: normal (free of EMN; *n* = 10), mild (*n* = 6) and severe EMN (*n* = 9). To explore the differences in proteins between groups, proteomic profiling and analysis were employed. Accordingly, 8241 annotated proteins out of 8725 total proteins were compared between normal and EMN groups and inspected based on differentially expressed proteins (DEPs). Through DEP analysis, 95 significant DEPs differed between normal and EMN groups. Among these DEPs, 41 significant proteins were categorised according to protein functions. Based on 41 significant proteins, 10 were involved in metabolism and 31 in non-metabolism. Interestingly, phospholipid phosphatase6 (PLPP6) and ATPase subunit alpha (Na+/K+-ATPase) were considered as potential proteins uniquely expressed in mild EMN and related to lipid and energy metabolism, respectively. Non-metabolism-related proteins (BRCA1, phosphorylase B kinase regulatory subunit: PHKA1, tyrosine-protein kinase receptor: ALK and rho-associated protein kinase: ROCK1) correlated to melanoma development differed among all groups. The results of our study provide a foundation for early EMN biomonitoring and prevention.

## 1. Introduction

Equine melanocytic neoplasm (EMN) is the cutaneous neoplasm [1], which occurs from uncontrolled melanin metabolism that affects melanoblast overproduction in the basal layer of the skin and the outer root sheath of hair follicles [2]. The genetic mutation of gene Syntaxin-17 (STX17) converts duplication, which is responsible for the vesicle transportation of malignant cell melanosomes and nuclei [3]. There is also an abnormality of the Agouti-signalling protein (ASIP), which contributes to an increase in melanocortin-1-receptor pathway signalling [3]. Equine melanocytic neoplasm can be found in tertiary tumours in horses susceptible to sarcoid and squamous cell carcinoma [1]. In addition, EMN has been identified as the genetic influence in grey coat in approximately 3–15% of all cutaneous neoplasms in horses [4]. According to several studies, there is a prevalence of EMN in grey horses [5] compared to horses with a more solid colour [6]. In this sense, EMN is considered a common neoplasm in grey horses. To observe EMN, typical lesions could be identified at specific locations, such as underneath the tail; in the perineum region; in the perianal region and at the muzzle, lip and eyelid [2]. Consequently, EMN was merely treated at late stage by surgery, immunotherapy and immune modulation or chemotherapy [1]. To overcome this, there is an urgent need for the early detection of EMN.

So far, there are few studies on candidate biomarkers for diagnostic EMN via the immunohistochemistry approach. However, such an approach could identify proliferation cell nuclear antigen (PCNA) and Ki-67 as candidate biomarkers for benign and malignant stages, which has been done in 27 grey and 4 solid-colour horses [7]. A receptor for activated C kinase 1 (RACK1) has also been identified immunohistochemically as a candidate biomarker for malignant stages in skin biopsies on grey horses [8]. Recently, proteomics has emerged as an alternative approach to identify and quantify proteins from grey horses at different EMN stages. However, there is no report on proteomic profiling of grey horses across different EMN stages.

In this context, this preliminary study investigated the proteomic profiling of grey horses across different EMN stages for further identification of key proteins playing functional roles in the prevalence of EMN. For this, we studied three stages, namely normal, mild and severe EMN. Protein analysis from serum samples was performed, followed by LC-MS/MS-based protein identification. The data were processed with different bioinformatics tools, using databases for protein quantitation and annotation. The obtained proteins were further classified into functional categories. To identify proteins that were uniquely expressed in mild or severe stages, the significantly differentially expressed proteins (DEPs), functional roles and functional categories between normal and EMN for mild or severe stages were then considered. Our results improve the understanding of genetics and pathophysiology pathways of EMN in horses, facilitating the identification of EMN-associated potential proteins for biomonitoring EMN prevention and treatment.

## 2. Materials and Methods

### 2.1. Animals and Serum Sample Collection

We used 25 grey horses from the Horseshoe Point International Riding School, Chonburi, Thailand. Physical examinations were performed in all horses, measuring heart rate, respiratory rate, gut sound, mucous membrane and rectal temperature. Afterwards, the region underneath the tail, the perineum region, along with the vulva or penis or scrotum, the perianal region, the area around the muzzle, both the upper and lower lips and the area around the eyelids were inspected for nodules of EMN; the findings were recorded for group classification based on the EMN lesion scoring system [9]. In this study, the classification levels 0, 1 and 2–5 were considered as control, mild and severe stages of EMN, respectively. Based on these criteria, 10 were randomly selected as the control group (normal without clinical sign of EMN; *n* = 10); horses with clinical signs of EMN at mild stage (*n* = 6) and severe stage (*n* = 9) were selected. The details on signalments, group classification and EMN levels of grey horses in this study are presented in Table 1. The experimental protocol used in this study was approved by the Institution of Animal Care and Use Committee of the Faculty of Veterinary Science, Kasetsart University, Bangkok, Thailand (ACKU63-VET-046).

### 2.2. Serum Sample Preparation

We collected 10 mL of serum samples from all horses from the jugular vein. The samples were kept in serum plain tubes at 4 °C for 30 min to coagulation and, subsequently, the serum was separated and stored at −80 °C until further analysis. Total serum protein was measured via the Lowry assay, using bovine serum albumin as standard [10]. Five µg of the protein sample was subjected to digestion. For this, the samples were diluted in 10 mM ammonium bicarbonate (AMBIC), and the disulphide bonds were reduced using 5 mM dithiothreitol (DTT) in 10 mM AMBIC at 60 °C for 1 h. Alkylation of sulfhydryl groups was performed by using 15 mM iodoacetamide (IAA) in 10 mM AMBIC at room temperature for 45 min in the dark. For digestion, protein samples were mixed with 50 ng/µL of sequencing-grade trypsin (1:20 ratio) (Promega, Walldorf, Germany) and incubated at 37 °C overnight. Prior to LC-MS/MS analysis, the tryptic peptides were dried and protonated with 0.1% formic acid before injection into LC-MS/MS.

### 2.3. Liquid Chromatography-Tandem Mass Spectrometry

The tryptic peptide samples were prepared for injection into an Ultimate3000 Nano/Capillary LC System (Thermo Scientific, Loughborough, UK), coupled to a HCTUltra LC-MS system (Bruker Daltonics Ltd.; Hamburg, Germany) equipped with a nano-captive spray ion source. Briefly, 5 μL of tryptic peptide samples were enriched on a µ-Precolumn 300 µm i.d. × 5 mm C18 Pepmap 100, 5 µm, 100 A (Thermo Scientific, Loughborough, UK), separated on a 75-μm I.D. × 15 cm and packed with Acclaim PepMap RSLC C18, 2 μm, 100 Å, nanoViper (Thermo Scientific, Loughborough, UK). The C18 column was enclosed in a thermostat column oven set to 60 °C. Solvents A and B, containing 0.1% formic acid in water and 0.1% formic acid in 80% acetonitrile, respectively, were supplied to the analytical column. A gradient of 5–55% solvent B was used to elute the peptides at a constant flow rate of 0.30 μL/min for 30 min. Electrospray ionisation was carried out at 1.6 kV using the CaptiveSpray. Nitrogen was used as a drying gas (flow rate about 50 L/h). Collision-induced-dissociation (CID) product ion mass spectra were obtained using nitrogen gas as a collision gas. Mass spectra (MS) and MS/MS spectra were obtained in the positive-ion mode at 2 Hz over the range of (m/z) 150–2200. The collision energy was adjusted to 10 eV as a function of the m/z value. The LC-MS analysis of each sample was performed in triplicate.

### 2.4. Quantification and Identification of Proteins

For the quantification of proteins, the DeCyder MS Differential Analysis software (DeCyderMS, GE Healthcare, Uppsala, Sweden) was used to quantify the proteins in individual samples; the Mascot search engine was applied to correlate MS/MS spectra to the Uniprot mammal database [11,12], which was accessed on 5 March 2020. We used Mascot’s standard settings: a maximum of three miss cleavages, a mass tolerance of 0.6 dalton for the main search, trypsin as digesting enzyme, carbamidomethylation of cysteine as fixed modification, oxidation of methionine as variable modification and the peptide charge state (1+, 2+ and 3+). Visualisation and statistical analyses were conducted using the MultiExperiment Viewer (MeV) in the TM4 suite software [13]. In the direction of protein identification, we selected the raw proteomic data from the horses for each group of normal, mild EMN and severe EMN stages. Maximum peptide intensities were log2-transformed in Microsoft Excel, providing the protein expression levels (PELs) for DEPs analysis. The raw MS/MS spectra data, protein sequences and PELs of the normal, mild EMN and severe EMN stages were deposited in the 8725-data repository. Raw MS/MS spectra data are available in ProteomeXchange: JPST001154 and PXD025793.

### 2.5. Comparative Differentially Expressed Protein (DEP) Analysis towards Functional Annotation

To compare the DEPs among normal, mild and severe EMN stages, we used the Wilcoxon rank-sum test and multiple testing via false discovery rate (FDR) correction, identifying significant proteins between the groups (adjusted *p*-value < 0.05). The significant protein expression was intended to serve as a biomarker to classify the normal and severity of EMN in horse. The Wilcoxon rank sum-test was selected in this study because it is a nonparametric test that compares medians between two groups of independent samples. For the functional annotation of DEPs, the KEGG database [14] was also used. The flowchart of proteomic profiling towards DEP analysis and functional annotation is shown in Figure 1.

## 3. Results

Twenty-five grey horses were classified according to the EMN lesion scoring system [9] and further differentiated into normal (free of melanoma: score 0), mild EMN (beginning stage: score 1) and severe EMN (prolong stage: score 2–4), based on the gross appearance of the mass and the typical site [5,15]. On the details regarding signalments, group classification and EMN grade are provided in Table 1. The EMN lesions were mainly presented in older horses (mild EMN 21.3 ± 5.72 and severe EMN 22.4 ± 5.77 years old) compared to younger ones (16.5 ± 4.17 years old).

The proteomic data from the 25 grey horses resulted in the identification of 173,760 independent mass spectral counts (i.e., 67,487 spectral counts for 10 normal samples, 43,300 spectral counts for 6 mild samples and 62,973 spectral counts for 9 severe samples). After partitioning of the samples to each group, the spectral counts per sample were presented between the normal stage (6749 spectral counts), EMN with mild stage (7216 spectral counts) and severe stages (6997 spectral counts), as shown in Table 2. After considering the unique protein sequences, 8241 annotated proteins out of 8725 total protein sequences were selected for further analysis. To explore the functions and pathways of proteome data of grey horses across EMN stages, DEP analysis towards functional annotation was performed.

A total of 8241 annotated proteins from the assessed proteomic data were included in the DEP analysis with the Wilcoxon rank-sum test and FDR correction. Focusing on the list of DEPs, 95 significant proteins were identified under an adjusted *p*-value < 0.05. Among these proteins, 41 significant proteins had categorised protein functions according to the KEGG database (Table 3). Among these, we found 10 proteins involved in metabolism and 31 proteins involved in non-metabolism categories, i.e., cellular processes (8 proteins), environmental information processing (7 proteins), genetic information processing (6 proteins), organismal systems (5 proteins) and human diseases (5 proteins). For non-metabolism categories, interestingly, we also found BRCA1, phosphorylase b kinase regulatory subunit (PHKA1), tyrosine-protein kinase receptor (ALK) and rho-associated protein kinase (ROCK1). According to the significant results for comparative protein expression levels (PELs), all groups are shown in Figure 2. To identify proteins that were uniquely expressed in mild or severe stages of EMN, the significant differentially expressed proteins (DEPs) for normal and mild stages were considered. Of 95 significant proteins, we found two significant proteins (PLPP6 and ATPase) in the majority of the samples of the mild stage and no expression in all samples from the normal group (Appendix A).

## 4. Discussion

### 4.1. Classification of the Horses and EMN Levels

Various researchers have reported that approximately 80% of EMN lesions occurred in grey horses older than 15 years [1,5,6]. On the one hand, the EMN lesion was mainly found in old grey horses. Furthermore, elderly greying is the crucial factor that tends to develop the EMN [6] and is caused by the effect of mitochondrial function by increased ATP production, with remarkable enhancement of lipid uptake metabolism progressed in the melanoma microenvironment surrounding and can aggressively promote tumour performance [16]. The most significant EMN lesion was found in the breed Lusitano (8/15), when compared to ponies (5/15) and thoroughbred horses (2/15), which is in agreement with the study of [17], where Lusitano was considered a predisposed breed for EMN. The greying of hair in horses represents a risk factor for EMN, and the Lusitano breed may have a higher L* (lightness or grey coat phenotype) value than other breeds [3]. The sex did not influence the incidence of EMN [18]. In contrast, there have been inconsistent reports regarding sex predilection. On study reported that multiple dermal melanoma (dermal melanomatosis) was more frequent in males [19]; however, in our study, we could not analyse this as we sampled a higher number of mares (19/25).

### 4.2. Analysis of Differentially Expressed Proteins and Functional Annotation of Grey Horses Across EMN Stages

Based on the results, the data from the analysis suggested that signalling and receptor protein pathways in the cell microenvironment are the key processes of equine and human melanoma development, indicating that to prevent tumour (or cancer) development, these processes need to be inhibited. Additionally, non-metabolic pathway are responsible for inducing rapid intra- and extra-cellular signalling responses to regulates cell proliferation, cell survival, growth responses, motility, invasion, survival, metabolism and gene transcription [20] via phosphorylation and dephosphorylation from kinase and phosphatase, respectively. These processes could be related to metabolic diseases in the cell; most cancer cells become heavily dependent on substrate level phosphorylation to meet their energy demands [21] through glucose and lipid metabolism pathways.

The BRCA1 is a tumour-suppressing gene encoding a large protein that is involved in numerous essential biological processes, including DNA damage repair, cell cycle control and transcriptional regulation [22]. Fragments of BRCA1 are involved in the pro-apoptotic program [23]. Experimental data have demonstrated that the expression of BRCA1 in the normal cell is related to the maintenance of genome integrity [22]. In contrast, deficiency of BRCA1 induced severe genome instability, eventually leading to tumorigenesis [24]. However, a high level of protein expression fragment of BRCA1 in the melanoma cell is indicated as being pro-apoptotic and continues or enhances the apoptotic program [23].

Anaplastic lymphoma kinase (ALK) is a transmembrane tyrosine kinase receptor that is frequently rearranged, mutated or amplified in specific neoplastic diseases, including lymphoma, neuroblastoma, lung cancer and, albeit to a lesser extent, melanoma [25]. In addition, ALK expression was not as specific as the phosphorylation enzyme mRNA that presented high significance in melanoma cell lines from a neuroectodermal origin [26]. Overexpression of ALK protein has been reported for many different types of cancer cell lines and human tumour samples [27]. Regardless of amplification, ALK overexpression is widely observed in nearly 100% of basal cell carcinoma [28] and in more than 50% of neuroblastoma, with only 10% of primary neuroblastoma [29]. It has been suggested that the oncogenic role of ALK is mediated via activation of tyrosine kinase, promoting the metabolism signalling cascade pathway of the melanoma and, sporadically, of other human cancer types, but not in normal tissues [26].

Like many other cancer types, the majority of rapidly proliferating melanoma cells metabolise glucose; when glycolysis is used at a high rate in proliferating tumour cells, it can supply the ATP necessary for survival while also supplying the material required for proliferation [30]. A comparative metabolic flux profiling of melanoma cell lines and normal melanocytes showed that all melanoma cells consumed more glucose as primary energy source, even in the presence of oxygen, and thus produced more lactate than melanocytes [31]. Lactate secretion cooperation to promote melanoma metastasis [32] via phosphorylation enzyme is involved in the pathway for PHKA1 [33] and lipid metabolism, as well as glucose metabolism for ROCK1. The PHKA1 catalyses the phosphorylation of serine [34], whereas ROCK1 catalyses that of serine/threonine in cancer development, progression and metastasis [35].

The PHKA1 has a low cancer and tumour cell-specificity. However, it is involved in the development of the melanoma cell [36], which consumes a high glucose level via phosphorylation when compared to the normal cells in this study. This leads us to infer that melanoma cells use large amounts of glycogen as an energy supply and store carbohydrates for melanoma progression. Elevated protein expression of ROCK1 and ALK has a strong prognostic value for cancers [36] in humans as well as EMN in equines. Our findings support the role of ALK, ROCK1 and PHKA1 as a protein kinase for activating the metabolism of the energy sources from macronutrients used by the melanoma to support growth, proliferation and survival. Consequently, metabolism-related pathways have acquired enormous relevance in cancer research [37]. It is speculated that other modifications in lipid metabolism are potential biomarkers in melanoma; however, the discovery of clinically useful biomarkers still requires the inclusion of consistent large-scale proteomic studies in clinical trials [38].

Lipid metabolism is associated with carbohydrate metabolism (glycolysis), as products of glucose, i.e., acetyl CoA, can be converted into lipids. Lipid metabolism refers to a complex set of molecular processes that include lipid uptake, de novo synthesis, transport and degradation [39]. Melanoma cells can adapt to the tumour microenvironment by using a variety of energy sources to maintain tumour growth and progression. Interestingly, there is evidence that numerous alterations of the lipid metabolic network can sustain cell growth and metastasis in melanoma cells [38], which increases the rate of lipogenesis, in which nutrient-derived carbon is converted into fatty acids (FAs), sterols and complex lipids that can fuel FA b-oxidation (FAO) in the mitochondria. Lipidogenesis mainly relies on the availability of acetyl-CoA [38]. In our study, most of the proteins were involved in lipid metabolism (7 out of 10 proteins); for example, lipin 2, phosphoinositide phospholipase C, beta-carotene oxygenase 1, long-chain fatty acid proteins, 3-hydroxy-3-methylglutaryl coenzyme A synthase, phospholipase D family member (PLD3) and sphingomyelin phosphodiesterase 3. These proteins were highly expressed in the mild and severe EMN stages, which could be an important target for treating several diseases, except for phospholipase D (PLD3). On our suggestion, the higher expression of PLD3 protein was presented in normal horses than in horse with mild or severe EMN. In addition, the expression of PLCH1 was found in obviously high values in mild and severe stages; this protein is involved in lipid metabolism in the microenvironment.

The upregulation of lipogenic enzymes has been described in various cancers [39], including melanoma. Increased lipid biosynthesis is thought to contribute to tumour cell proliferation, partly by providing large amounts of “building materials” for cell membrane production and energy for b-oxidation. An increase in the “lipid second messenger” molecules mediates oncogenic pathways [39], including phosphatidylinositol bisphosphate (PIP2), diacylglycerol (DAG) and inositol triphosphate (IP3) by lipogenesis [34] with phospholipase C [40]. Additionally, 3-hydroxy-3-methylglutaryl coenzyme A synthase, as an enzyme, condenses acetyl-CoA with acetoacetyl-CoA. Both, acetyl-CoA and acetoacetyl-CoA are precursors for FA and cholesterol synthesis [38]. Cholesterol, FA and sphingolipids, i.e., sphingomyelin, are part of the lipid raft, acting as signalling hubs in cancer proliferation, adhesion and migration [38]. In this study, lipidogenesis homeostasis was dysregulated in cancer cells, and changes in FA and cholesterol metabolism substantially impacted EMN progression, including invasion. The enzymes EVOL2, LPIN and SMPD3 are involved in lipid metabolism, whereas BCO1 activity may be a more meaningful indicator of circulating cholesterol concentration [41].

### 4.3. Identification of Uniquely Expressed Proteins of Mild-Stage EMN

Interestingly, the significant different expression of two proteins was discovered between normal and mild stages of EMN, which are represented in Appendix A. These proteins were involved in metabolism, specifically lipid metabolism, i.e., phospholipid phosphatase 6 (PLPP6, PEL of 13.57), and energy metabolism, i.e., sodium/potassium-transporting ATPase subunit alpha (Na+/K+-ATPase, PEL of 14.68). The protein PLPP6 is involved in the inhibitory action of lipoxins on superoxide production in neutrophils as well as DAG and IP3 signalling. Generally, lipoxins can control the proliferation of immune cells and cancer cells and represent a unique class of lipid mediators that can function as “braking signals” in inflammation. Once lipoxin is inhibited, leading to inflammation, cancer cells may occur. Several studies report that plasma measurement of lipoxins may be used to predict prognosis and responses to therapy. This could also play a role in other auto-immune diseases. Taken together, PLPP6 could be used as a potential protein for the prognosis of mild stages of EMN. Na+/K+-ATPase is an enzyme encoded by the ATP1A1 gene and is the integral membrane protein responsible for establishing and maintaining the electrochemical gradients of Na and K ions across the plasma membrane. The PLPP6 has been known as polyisoprenyl diphosphate phosphatase 1 (PDP1) or candidate sphingomyelin synthase type 2b (Css2b), hydrolysing presqualene diphosphate (PSPD), farnesyl diphosphate (FDP), phosphatidase (PA), lysophosphatidase (LPA) and sphingosine 1-phosphate. The PSPD is directly inhibited by PLD [42], indicating that, in our study, PLD3 expression is lower in mild and severe stages than in the normal stage. Additionally, sphingomyelin synthase is the last enzyme for sphingomyelin synthesis, and therefore, the activity of sphingomyelin synthase should directly affect the sphingomyelin levels in cells and in circulation [43]. We observed high expression levels of enzymes sphingomyelin phosphodiesterase 3 (SMPD3) in the mild stage compared to the other stages. The SMPD3 catalyses sphingomyelin to ceramide, and the degradation of ceramide into sphingosine is associated with melanoma progression [44].

## 5. Conclusions

Based on the preliminary results, we propose that the metabolic state of EMN dictates non-metabolism protein expression, i.e., of phosphorylase b kinase regulatory subunit (PHKA1), tyrosine-protein kinase receptor (ALK) and rho-associated protein kinase (ROCK1), affecting signalling responses to regulate cell proliferation, cell survival, growth responses, motility, invasion, survival, metabolism and gene transcription via phosphorylation of the EMN microenvironment for the development and use of primary energy sources (sugar). However, EMN cells can use a variety of energy sources to maintain tumour growth and progression. Thus, EMN cells increase the rate of lipid metabolism for progression. Lipid metabolism may offer novel potential proteins for EMN levels, i.e., PLCH1, HMGCS2 and PLPP6. The proteomic profiling achieved in this study revealed potential proteins for the biomonitoring of early EMN, facilitating rapid prevention and treatment.

## Figures and Tables

**Figure 1 animals-11-01913-f001:**
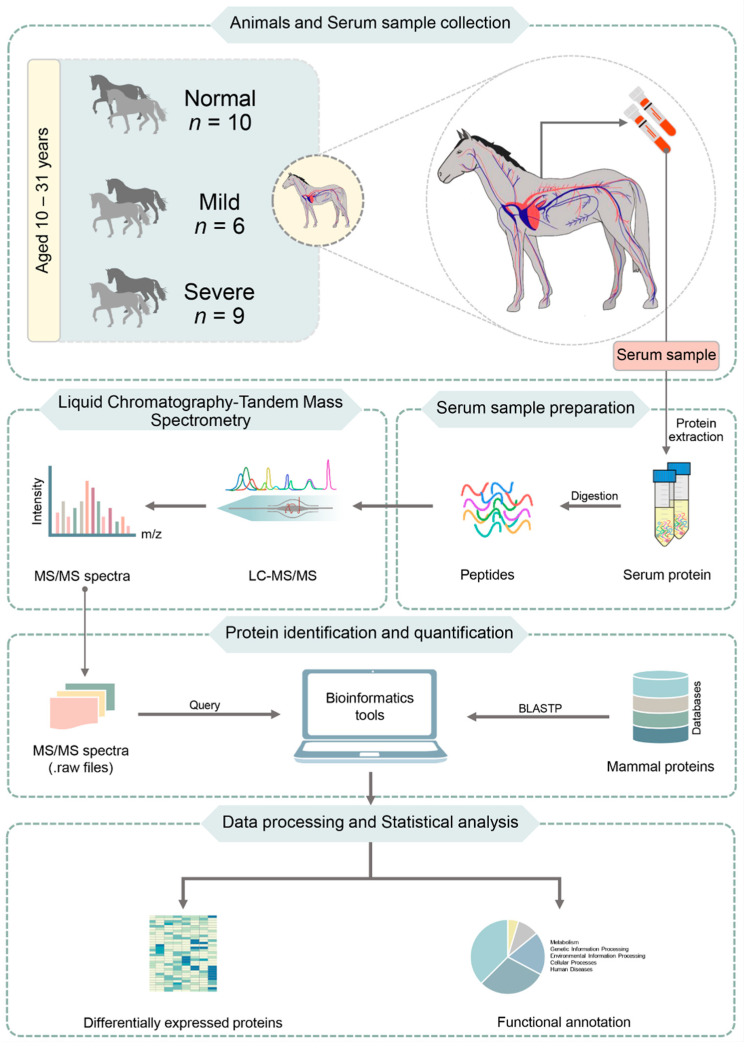
Flowchart of proteomic profiling towards DEP analysis and functional annotation. BLASTP (Protein BLAST) compares one or more protein query sequences to subject protein sequences in database. It allows to find regions of local similarity between protein sequences.

**Figure 2 animals-11-01913-f002:**
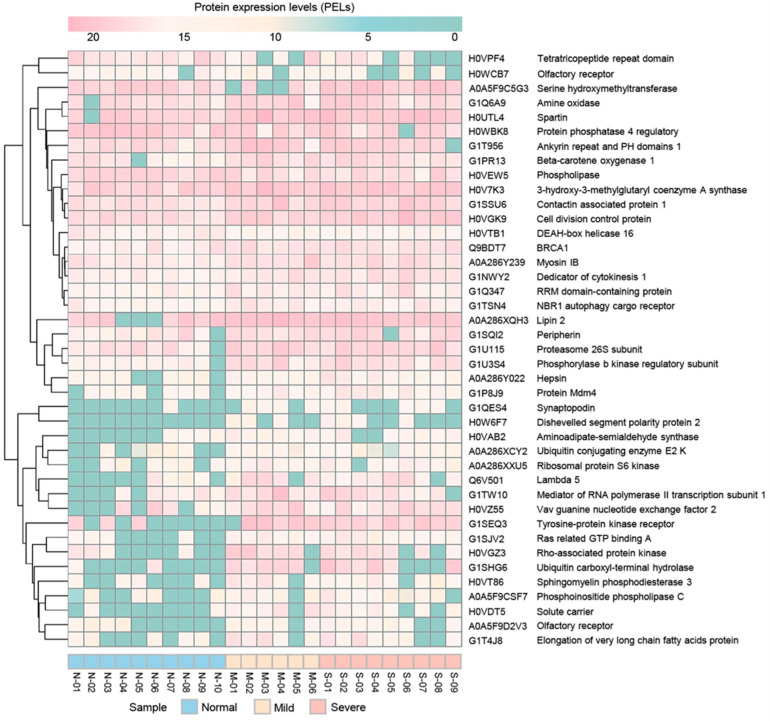
Results for comparative protein expression of the groups with normal, mild and severe EMN. The heatmap was generated using the pheatmap package (https://cran.r-project.org/web/packages/pheatmap/pheatmap.pdf accessed on 4 January 2019), applied in R (version3.5.3, R Foundation for Statistical Computing, Vienna, Austria.) (http://www.R-project.org accessed on 4 January 2019) for conception. N, M and S represent normal, mild and severe EMN, respectively.

**Table 1 animals-11-01913-t001:** Signalments, group classification and EMN levels of grey horses.

ID	Age (Years)	Sex	Breed	Colour	Weight (kg)	EMN Level ^1^
Normal grey horses (*n* = 10)						
1	12	Mare	Lusitano	Grey	506	0
2	13	Mare	Lusitano	Grey	560	0
3	13	Mare	Lusitano	Grey	447	0
4	15	Mare	Pony	Grey	270	0
5	15	Mare	Lusitano	Grey	425	0
6	15	Mare	Thoroughbred	Grey	470	0
7	18	Mare	Lusitano	Grey	485	0
8	19	Mare	Lusitano	Grey	470	0
9	19	Mare	Lusitano	Grey	425	0
10	26	gelding	Pony	Grey	221	0
Grey horses with mild EMN group (*n* = 6)						
11	15	Gelding	Lusitano	Grey	470	1
12	18	Mare	Lusitano	Grey	421	1
13	18	Mare	Lusitano	Grey	476	1
14	22	Mare	Pony	Grey	307	1
15	24	Mare	Lusitano	Grey	420	1
16	31	Mare	Pony	Grey	260	1
Grey horses with severe EMN group (*n* = 9)						
17	10	Gelding	Thoroughbred	Grey	432	2
18	17	Gelding	Lusitano	Grey	461	2
19	22	Mare	Pony	Grey	275	3
20	23	Mare	Lusitano	Grey	532	4
21	23	Mare	Thoroughbred	Grey	506	2
22	26	Mare	Pony	Grey	280	2
23	26	Mare	Lusitano	Grey	432	3
24	26	Stallion	Pony	Grey	191	2
25	29	Stallion	Lusitano	Grey	412	2

^1^ The EMN lesion scoring system based on the report of [9].

**Table 2 animals-11-01913-t002:** Assigned spectral counts and unique protein sequences.

	Protein ID Annotation		
EMN Stages	With Annotated Protein IDs ^1^	Without Annotated Protein IDs	Total
	Total Spectral Counts ^2^ (Spectral Counts Per Sample)		
N	63,795 (6380)	3692 (369)	67,487 (6749)
M	40,922 (6820)	2378 (396)	43,300 (7216)
S	59,519 (6613)	3454 (384)	62,973 (6997)
	Total Protein Sequences ^2^		
N	8187	481	8668
M	8194	484	8678
S	8231	484	8715
	Total Unique Protein Sequences ^3^		
	8241	484	8725

Notes: N, M, S represent normal, mild and severe stages of EMN, respectively. ^1^ Annotated protein IDs are based on a protein ID with assigned function from the Uniprot database. ^2^ Spectral counts are the number of spectra assigned to a peptide sequence. ^3^ Unique protein sequences can be assigned to a single protein.

**Table 3 animals-11-01913-t003:** List of 41 significant proteins and associated functions across EMN stages.

Protein ID	Protein Function	PELs ^a^		
		N	M	S
Metabolism				
A0A286XQH3	Lipin 2 (LPIN2)	19.08	20.44	20.10
A0A5F9C5G3	Serine hydroxymethyltransferase (SHMT1)	19.34	8.49	20.07
A0A5F9CSF7	Phosphoinositide phospholipase C (PLCH1)	3.28	13.59	14.48
G1PR13	Beta-carotene oxygenase 1 (BCO1)	16.94	18.34	18.00
G1Q6A9	Amine oxidase (LOC102438245)	18.34	19.09	19.09
G1T4J8	Elongation of very long-chain fatty acids protein (ELOVL2)	6.15	16.18	15.04
H0V7K3	3-hydroxy-3-methylglutaryl coenzyme A synthase (HMGCS2)	19.20	20.00	20.04
H0VAB2	Aminoadipate-semialdehyde synthase (AASS)	0.00	15.27	15.91
H0VEW5	Phospholipase D family member (PLD3)	19.33	18.36	18.04
H0VT86	Sphingomyelin phosphodiesterase 3 (SMPD3)	0.00	16.27	14.77
Non-metabolism				
A0A286XCY2	Ubiquitin conjugating enzyme E2 K (UBE2K)	4.64	14.86	13.73
G1Q347	RRM domain-containing protein (LOC102432793)	16.23	17.23	17.09
G1U115	Proteasome 26S subunit, non-ATPase 6 (PSMD6)	17.27	19.07	18.79
H0VPF4	Tetratricopeptide repeat domain 37 (TTC37)	17.70	16.10	12.63
H0VTB1	DEAH-box helicase 16 (DHX16)	16.80	15.71	15.83
Q9BDT7	BRCA1 (Fragment)	16.73	17.77	17.90
A0A286XXU5	Ribosomal protein S6 kinase (RPS6KA4)	11.98	14.80	16.06
G1SJV2	Ras related GTP binding A (RRAGA)	0.00	16.25	15.88
G1SSU6	Contactin-associated protein 1 (CNTNAP1)	18.04	18.78	19.19
G1U3S4	Phosphorylase b kinase regulatory subunit (PHKA1)	16.30	18.29	18.33
H0VGZ3	Rho-associated protein kinase (ROCK1)	0.00	18.55	17.69
H0VZ55	Vav guanine nucleotide exchange factor 2 (VAV2)	6.71	18.23	18.10
H0W6F7	Dishevelled segment polarity protein 2 (DVL2)	0.00	5.82	0.00
G1NWY2	Dedicator of cytokinesis 1 (DOCK1)	16.49	17.43	17.94
G1P8J9	Protein Mdm4 (MDM4)	14.81	15.93	16.69
G1QES4	Synaptopodin (SYNPO)	0.00	14.42	13.94
G1SHG6	Ubiquitin carboxyl-terminal hydrolase (USP2)	0.00	18.90	18.04
G1T956	ArfGAP with coiled-coil, ankyrin repeat and PH domains 1 (ACAP1)	18.29	19.61	19.34
G1TSN4	NBR1 autophagy cargo receptor (NBR1)	15.87	17.05	16.63
H0UTL4	Spartin (SPART)	19.06	20.14	19.83
H0VGK9	Cell division control protein (CDC6)	18.25	19.45	19.51
A0A5F9D2V3	Olfactory receptor (OR3A1)	0.00	13.98	14.82
G1TW10	Mediator of RNA polymerase II transcription subunit 1 (MED1)	15.49	18.02	17.50
H0VDT5	Solute carrier family 17 member 8 (SLC17A8)	0.00	16.77	15.71
H0WBK8	Protein phosphatase 4 regulatory subunit 3C (PPP4R3C)	20.06	19.29	19.15
H0WCB7	Olfactory receptor family 1 subfamily I member 1 (OR1I1)	15.52	14.55	11.49
A0A286Y022	Hepsin (HPN)	13.13	14.67	16.06
A0A286Y239	Myosin IB (MYO1B)	17.18	17.98	17.87
G1SEQ3	Tyrosine-protein kinase receptor (ALK)	0.00	20.00	19.52
G1SQI2	Peripherin (PRPH)	16.57	18.21	18.81
Q6V501	Lambda5 (IGLL1)	0.00	17.69	16.59

Notes: N, M, S represent normal, mild and severe stages of EMN, respectively. **^a^** Median values of PELs are presented.

## Data Availability

Raw MS/MS spectra data are available in ProteomeXchange: JPST001154 and PXD025793.

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
