# Peer review of "Preliminary Study: Proteomic Profiling Uncovers Potential Proteins for Biomonitoring Equine Melanocytic Neoplasm"

_animals, 2021, doi:10.3390/ani11071913_

Round 1

Reviewer 1 Report

Proteomic investigation on serum of grey horses was performed in different stages of the disease, in order to find markers for biomonitoring and prevention of euqine melanocytic neoplasm.
I would not call blood collection „non-invasive“, although – agreed – this is less invasive than surgery.

Line 81: there is no need to „extract“ proteins from serum, as they are the main components and nicely dissolved. Please rephrase, e.g. „protein analysis of serum samples...“

Please refer to Table 1 already in the section 2.1.

Line 97: measuring? (mearing)

Line 109: „isolated“ is not appropiate. Replace by „Total serum protein was measured...“

Line 112: Serum proteins are already dissolved. Replace by „..samples were diluted in 10 mM...“

Please give version of UniProt or date of access.

Did you use any cutoff for protein identification (number of peptides; presence in e.g. all technical replicates or in ¾ of all biological replicates)?

Raw data have been deposited on PRIDE, but no reviewer login details are given and thus cannot be accessed. Data quality cannot be evaluated.

Lines 174 to 177 do not fit into the results section.

Lines 207/208 and section 4.3: You mention that two proteins were significantly different in the N/M comparison, but do not name them in section 3. If those are PLPP6 and an ATPase, as mentioned in section 4.3 why did I not find them in Table 3 and Figure 2? Did I overlook them or do they have an alternative name? Please explain.

Lines 224-227: are the numbers referring to the present study (but they do not correspond to the description) or the reference? Please rephrase accordingly.

Line 314: lipogenesis? (ipogenesis)

Did you attempt to validate any of your findings with an independent method? Would you please check with similar diseases (either cancer or skin diseases) how unique your findings are or whether you pick up similar candidates?

Please proof-read to eliminate typos.

Author Response

Dear Reviewer 1

Thank you for your kindness suggestion. We carefully corrected the revised manuscript following your suggestion. The correction was presented in the revised manuscript with track changed and also in this document.

Based on the editor’s suggestion. The editor need to:

  1. cite the references and reduce the similarity;

We reduced the number of references and similarity following your suggestion.

  1. add copy right for "Desser et al. (1980)" under Table 1;

Response: We have corrected to reference “Desser et al. (1980)” under Table 1 following the author guideline.

  1. add statement regarding to "prelimiary study".

Response: We provided the statement of preliminary study in the Title and also in the revised manuscript following your suggestion.

If you have any question or suggestion, please do not hesitate to contact us. We would like to answer and correct to you as soon as possible.

Best regards,

Attawit Kovitvadhi & Sittiruk Roytrakul

Corresponding authors

Reviewer 2 Report

The paper is well-organized and interesting article from Tesena et. al.  Overall, this paper is of novel in equine area. Their conclusion is clear. The authors explored a way to track the status or development of EMN. Although I put forward some questions, this is an interesting paper. However, some raised questions need to be concerned. Overall, this project is well-designed, and the results support their conclusion. Importantly, some raised problems need to be addressed.

  • It is necessary that the authors should add some description or content in background, their significance, and advance.

e.g, if similar method was employed other species,

  • In research of big animals, the sample size used in this study is not enough.
  • This paper needs some editorial work respect to style and grammar.
  • Statistic method section, their description is not clear and I am confused of what statistic methods were used.
  • In discuss section, they should improve their discussion deeper including application.

Author Response

Dear Reviewer 2

Thank you for your kindness suggestion. We carefully corrected the revised manuscript following your suggestion. The correction was presented in the revised manuscript with track changed and also in this document.

Based on the editor’s suggestion. The editor need to:

  1. cite the references and reduce the similarity;

We reduced the number of references and similarity following your suggestion.

  1. add copy right for "Desser et al. (1980)" under Table 1;

Response: We have corrected to reference “Desser et al. (1980)” under Table 1 following the author guideline.

  1. add statement regarding to "prelimiary study".

Response: We provided the statement of preliminary study in the Title and also in the revised manuscript following your suggestion.

If you have any question or suggestion, please do not hesitate to contact us. We would like to answer and correct to you as soon as possible.

Best regards,

Attawit Kovitvadhi & Sittiruk Roytrakul

Round 2

Reviewer 1 Report

The authors have made some of the suggested amendments. Thank you for adding Table S1.

Please additionally correct the following:
a) In section 2.1 cite Table 1 (in addition to the already existing citation on the Results section)

  1. b) Please delete the first sentence on page 6, this does not fit in there and nothing to do with your found results.
  2. c) The first paragraph of 4.1. needs English refinement in several places.
  3. d) I do not understand the sentence “We suggest that PLD3 protein, more expressed than mild and severe stages, indicate that EMN using PLCH1 for lipid metabolism in the microenvironment, both enzymes have is an important target for treating several diseases, including cancers [43].“ While the first part should probably be “We suggest that PLD3 protein, more expressed in normal horses than in mild and severe stages, …”, I do not understand the second half of this sentence.
  4. e) Section 4.3. “According to…?”
  5. f) page 12, line 4: instead of “indicating that” you may mean “explaining why in our study…”
  6. g) Reference 24: the link does not lead to the indicated paper.

Just two comments:

  1. a) You have given a link to a site for download of your data (which is fine). However, my request was for the reviewer login details that have been sent to you from the repository. The repository contains also a description of your experiment and settings.
    b) To my question about validation of some of your findings, you did not answer properly. Statistical evaluation is fine, but for really good evaluation and confirmation of proteomic results, additional tests would be necessary. A validation would be testing for some of the proteins you found differentially regulated with an independent method, e.g. by Western blot, or confirming the affected pathways by metabolomics testing.

Author Response

Dear Reviewer 1

Thank you for your kindness suggestion. We carefully corrected the revised manuscript following your suggestion. The correction was presented in the revised manuscript and also in attached document. If you have any question or suggestion, please do not hesitate to contact us. We would like to answer and correct to you as soon as possible.

Best regards,

Attawit Kovitvadhi
